# A Novel CoviddDetNet Deep Learning Model for Effective COVID-19 Infection Detection Using Chest Radiograph Images

**Naeem Ullah** [1], **Javed Ali Khan** [2,*], **Sultan Almakdi** [3,*], **Mohammad Sohail Khan** [4], **Mohammed Alshehri** [3], **Dabiah Alboaneen** [5] **and Asaf Raza** [1]

1   Department of Software Engineering, University of Engineering and Technology, Taxila 47050, Pakistan; naeemullahfeb1997@gmail.com (N.U.); asafraza94@gmail.com (A.R.)
2   Department of Software Engineering, University of Science and Technology Bannu, Bannu 28100, Pakistan
3   Department of Computer Science, College of Computer Science and Information System, Najran University, Najran 55461, Saudi Arabia; msalshehry@nu.edu.sa
4   Department of Computer Software Engineering, University of Engineering and Technology Mardan, Mardan 23200, Pakistan; sohail.khan@uetmardan.edu.pk
5   Computer Science Department, College of Sciences and Humanities, Imam Abdulrahman Bin Faisal University, Jubail 31961, Saudi Arabia; dabuainain@iau.edu.sa
*   Correspondence: engr_javed501@yahoo.com or javed_ali@ustb.edu.pk (J.A.K.); saalmakdi@nu.edu.sa (S.A.)

**Abstract:** The suspected cases of COVID-19 must be detected quickly and accurately to avoid the transmission of COVID-19 on a large scale. Existing COVID-19 diagnostic tests are slow and take several hours to generate the required results. However, on the other hand, most X-rays or chest radiographs only take less than 15 min to complete. Therefore, we can utilize chest radiographs to create a solution for early and accurate COVID-19 detection and diagnosis to reduce COVID-19 patient treatment problems and save time. For this purpose, CovidDetNet is proposed, which comprises ten learnable layers that are nine convolutional layers and one fully-connected layer. The architecture uses two activation functions: the ReLu activation function and the Leaky Relu activation function and two normalization operations that are batch normalization and cross channel normalization, making it a novel COVID-19 detection model. It is a novel deep learning-based approach that automatically and reliably detects COVID-19 using chest radiograph images. Towards this, a fine-grained COVID-19 classification experiment is conducted to identify and classify chest radiograph images into normal, COVID-19 positive, and pneumonia. In addition, the performance of the proposed novel CovidDetNet deep learning model is evaluated on a standard COVID-19 Radiography Database. Moreover, we compared the performance of our approach with hybrid approaches in which we used deep learning models as feature extractors and support vector machines (SVM) as a classifier. Experimental results on the dataset showed the superiority of the proposed CovidDetNet model over the existing methods. The proposed CovidDetNet outperformed the baseline hybrid deep learning-based models by achieving a high accuracy of 98.40%.

**Keywords:** chest X-ray; COVID-19; classification; detection; deep learning models

## 1. Introduction

The coronavirus family has hundreds of virus types; although only seven are harmful to humans [1]. Mammals such as bats transfer these viruses into the human body [2]. The virus can transfer from one animal (or person) to another through the air and by physical interaction with a COVID-19 positive patient, i.e., handshaking [3]. The COVID-19 virus causes an acute respiratory infection in people, and it has now become the world's greatest pandemic. At first, the virus contaminated residents of Wuhan, China, in December of 2019 [4]. The COVID-19 virus is lethal to humans due to its rapid propagation. The World Health Organization (WHO) has officially declared that COVID-19 is a pandemic [5,6]. COVID-19 indicators are high temperature, exhaustion, coughing, loss of taste, breathing

difficulty, etc. [7]. The virus usually affects the lungs in humans, producing pneumonia in extreme situations. As a result, the oxygen level in the body plummets. COVID-19 takes 3 to 13 days to grow, but symptoms in the patient's body take 6 to 7 days to appear. So far, the WHO has documented 396,558,014 COVID-19 cases globally, with 5,745,032 losses [8]. According to the Pakistani government, 1,531,242 cases of COVID-19 have been reported in Pakistan, with 30,381 fatalities and 1,498,017 recoveries [9]. As a response to the unpredicted COVID-19 outbreak, numerous research centers' research and innovation groups are working hard to create an accurate detection method and treatment vaccine [10]. Researchers from various fields, such as machine learning, computer science, and artificial intelligence, collaborate to control and mitigate the epidemic by sharing their technical insights and alternative solutions [11–15].

Moreover, existing COVID-19 detection tests are inefficient and require several hours to produce results. The antibody and polymerase chain reaction (PCR) test is usually used to identify COVID-19 around the globe. The antibody test, which is an indirect method of testing, can determine if the immune system has come into contact with the virus. Antibodies can take up to 9 to 28 days to form after an infection has taken hold, which is a long time. If the diseased person is not kept separate, the infection can spread. In most cases, PCR tests are used in medical research [16,17]. However, the number of patients continues to rise, and performing enough PCR testing has become difficult due to time constraints, a lack of medical resources, and the associated costs [18–20]. A fundamental constraint of PCR tests is the high expense of importing the required chemicals and other ingredients necessary in the kits. One PCR test costs almost up to USD 30 (compared to Pakistani rupees), and the price varies depending on availability in different regions of the world. That takes us to the following constraint: availability; not every country has the exact needs and resources. Some states have a larger population and fewer kits available, while others have more kits than are needed. As a result of COVID-19, clinical laboratories have developed, tested, and implemented various virus detection approaches. It has been crucial in identifying patients, making isolation recommendations, and assisting with disease control. As the need for COVID-19 testing has expanded, laboratory professionals have run across a growing number of barriers, doubts, and, in some cases, disagreements. As a result, there is an urgent need to develop alternative testing (automated COVID-19 diagnosis) techniques that can reliably detect the virus in a short period and low cost, allowing patients to be identified and quarantined or isolated quickly. One of the alternative solutions is to use chest radiographs for COVID-19 detection. One radiograph costs approximately USD 3; therefore, we can obtain a large number of image samples to efficiently and correctly identify COVID-19 using chest radiograph images. Furthermore, the time required to conduct a chest radiograph is approximately 15 min compared to the PCR test.

An important aspect of home tests for COVID-19, which are relatively simple to perform and interpret, is the immediacy of test results, between 10 and 30 min. Previously, patients had to wait several days or more for results from commercial reference laboratories. There are several important issues regarding using and interpreting home COVID-19 tests. Foremost among these are obtaining a quality specimen and the performance of the test. The process for the consumer has been simplified by the manufacturer-provided visual aids, videos, or online guidance to assist in specimen collection and understanding test performance. For home emergency use, authorization from the Food and Drug Administration (FDA) required feasibility data showing that people in the authorized age ranges can safely and accurately perform these tests. Individuals performing these tests must read and follow the manufacturer's instructions. Regardless of the test type, whether performed at home or in the laboratory, an inferior quality specimen often translates into inferior test results. Furthermore, positive results in asymptomatic individuals are less accurate and should be confirmed by more accurate tests. Additionally, home-based COVID-19 testing is not common globally, especially in underdeveloped countries. For example, currently, there is not a home-based COVID-19 testing facility in Pakistan. In contrast, the PCR-based

COVID-19 facility is frequently available globally but is time-consuming compared to the proposed approach.

Furthermore, COVID-19 has been reliably detected in its initial stages using a variety of medical imaging modalities, including chest radiography, electrocardiogram trace images, and computed tomography (CT) scan. Medical practitioners place a higher value on chest radiograph images since they are easily accessible through radiology departments. According to radiologists, chest radiograph images aid in the precise knowledge of chest pathology. As a result, the X-ray modality [21] is the first low-cost and low-risk approach for the COVID-19 analysis. The X-ray approach is an extensively utilized, most effective and accessible tool to identify and diagnose pneumonia [22]. Pneumonia is an infection that inflames the lungs' air sacs. X-ray serves an important role in clinical care and epidemiological studies. Detecting pneumonia in chest X-rays, on the other hand, is a difficult task that requires the presence of professional radiologists. We offer a model that can detect pneumonia from chest X-rays more accurately than experienced radiologists in this paper. X-rays are simpler, quicker, inexpensive, less harmful, and expose individuals to less radiation than CT and magnetic resonance imaging (MRI) [23].

Recent research has shown that Artificial Intelligence (AI) methods employing deep learning (DL) methodologies can identify numerous disorders on chest radiographs with accuracy comparable to expert radiologists [24–27]. When experienced radiologists are unavailable, these computer-aided detection (CADs) systems can help with imaging-based patient classification in resource-constrained situations and increase practitioner's chest radiograph interpretation accuracy and inter-reader variability [28,29]. Deep learning-based CAD has been shown in several recent studies to detect and identify COVID-19 on chest radiographs with high (radiologist-level) accuracy and to use it in medical practice [30,31]. DL approaches employ unstructured data, extract more significant features automatically [32], and generate more accurate results than classic ML techniques. Several investigations were started at the beginning of 2020 to build automated DL models for reliable COVID-19 detection [33]. Convolutional neural networks (CNNs) were employed in most of this research to classify and assess COVID-19-infected or normal chest X-ray pictures. To detect and identify brain tumors from magnetic resonance images (MRI) data, specialists can use CAD based on classical DL [34]. CNNs are commonly used in image classification and identification applications such as MRI brain cancer image classification [35] and others.

According to our knowledge, when it comes to AI-driven tools that use imaging techniques, COVID-19 does not have a lot of state-of-the-art literature. Existing COVID-19 detection research has certain limitations, i.e., low accuracy of COVID-19 detection. Most studies relied on datasets with fewer images (small datasets). There are fewer training data, the model is not perfectly generalized, and the training samples may have been overfitted. Most studies use classic ML and transfer learning algorithms to detect and assess COVID-19. Still, the most concerning limitation of traditional ML (such as support vector machine (SVM)) is the long training time for large datasets. In contrast, the most concerning limitations in transfer learning systems are negative transfer and overfitting. One of the drawbacks of using pre-trained classification approaches is that they are frequently trained on the ImageNet database, which contains images that are unrelated to medical images. As a result, putting in place effective CADS to reliably and quickly identify COVID-19 from chest radiographs remains a difficult task. To address these limitations, the CovidDetNet DL model is proposed. It utilizes filter-based feature extraction, which can help achieve high classification performance. The proposed model has been created with a convolutional layer and both ReLu and Leaky ReLu activation functions, which extract the most detailed and important features from the chest radiograph images. The architecture can minimize many weight parameters by using a max-pooling operation. We implemented both Relu and leaky ReLu activation functions and batch and cross channel normalization operations in the proposed model, making it a novel COVID-19 detection and classification method. The proposed approach also tackles the issue of PCR kit scarcity by requiring only an X-ray machine, which is currently found in most hospitals across the world. As a result, people

will not have to wait longer for colossal PCR kit shipments. The proposed approach would efficiently enable the contact and isolation of COVID-19 individuals and limit community transmission with the rapid detection of COVID-19. Therefore, as chest radiographs are low-cost and time-efficient, and available in practically every clinic, chest radiograph images were utilized as a sample dataset in this study. The suggested architecture was validated using a standard Kaggle (publicly available) dataset. According to the results, the proposed structure performs satisfactorily in COVID-19 detection in test accuracy. The main contributions of this research are:

- A CovidDetNet DL model is proposed to detect COVID-19 positive cases using chest radiograph images.
- Chest radiographs are favored over CT scans since X-ray machines are readily available in most hospitals and CT scans emit less ionizing radiation.
- To evaluate the efficacy of the proposed model, we compared the proposed CovidDetNet model performance with hybrid approaches on the same dataset and experimental configuration. For this purpose, we utilized various classification metrics, i.e., accuracy, precision, recall, and f1-score.
- We evaluated the performance of the proposed novel CovidDetNet DL model on a standard COVID-19 radiography database.

The remainder of this paper is arranged as follows. Section 2 presents the related work about COVID-19 detection methods. The motivation and description of the proposed work are discussed in Section 3. Section 4 explains the datasets, evaluation measures, and experimental results. Section 5 discusses the proposed approach and explains the future directions and Section 6 concludes the paper.

## 2. Related Work

Although COVID-19 has only just begun to spread, researchers have created a substantial number of research approaches in such a short time. To detect and classify COVID-19 images, which is still under research and needs further improvements, many ML, hybrid, and DL approaches have been presented in the literature. To highlight the critical literature work in COVID-19 detection and classification, below we discuss and analyze several related works on COVID-19.

Mahdy et al. [36] used multi-level thresholding and an SVM approach to identify and detect COVID-19. They used a median filter to increase the input image contrast after analyzing the patient's lung radiograph image. The Otsu objective function is then utilized to create a multi-level picture segmentation threshold. Next, the SVM was employed to distinguish between diseased and uninfected lungs. Using the proposed model, the authors achieved high average accuracy for classifying lung radiograph images than the existing approaches. Singh et al. [37] introduced a research method based on the least-squares SVM (LS-SVM) and autoregressive integrated moving average (ARIMA) to detect and identify COVID-19. The study's research data were obtained from the US, UK, Italy, France, and Spain, where most confirmed coronavirus cases occurred. The authors used different feature extraction methods to increase the performance of the proposed approach. The dataset is then fed into the algorithm that predicts the disease's spread one month ahead of time. LS-SVM outperformed ARIMA in terms of accuracy. Based on the locality-weighted learning and self-organization map (LWL-SOM), Osman et al. [38] proposed a novel COVID-19 detection method. They employed the SOM approach to group pictures from the chest radiographs dataset based on similar features in distinct clusters to distinguish between COVID-19 and normal patients. The LWL approach is then used to create a model for detecting COVID-19. The proposed model improved the correlation coefficient performance outcomes between COVID-19, normal, and pneumonia cases; pneumonia and normal cases; COVID-19 and pneumonia cases; and COVID-19 and normal cases. The proposed model improved the correlation coefficient performance outcomes. Current ML-based methods that use AI evaluation measures to distinguish COVID-19 and normal patients outperformed the proposed model. In [39], the authors

present a fusion scheme based on an ML system using three significant texture features, namely, local binary pattern (LBP), fractal dimension (FD), and grey level co-occurrence matrices (GLCM). In experimental results, to demonstrate the efficiency of the proposed scheme, we have collected 300 CT scan images from a publicly available database.

Furthermore, traditional ML approaches perform poorly compared to DL techniques because they rely heavily on manual feature extraction and adequate feature selection. In contrast, DL approaches use unstructured data, extract more robust deep features, and produce more accurate results than traditional ML algorithms. Recently, DL algorithms have been frequently employed to extract classification features automatically. Classifiers based on DL can be utilized to develop fully automated classifiers that can detect COVID-19 using chest radiograph images. Recently, Ozturk et al. [40] proposed a binary classifier that recognizes COVID-19 and normal chest radiograph pictures and a multiclass classifier that detects COVID-19, pneumonia, and normal images using the DarkNet transfer learning (TL) architecture. Similarly, the Xception TL approach was utilized as a pre-trained network by Khan et al. [41]. The tests were carried out on publicly available datasets. Apostolopoulos et al. [42] used the MobileNet DL algorithm, trained the model from scratch, and extracted various classification features for COVID-19 classification. On the COVID-19 diagnostic, Ucar et al. [43] used the Bayesian optimization method to optimize the SqueezeNet network. The COVID-19 images dataset has also been improved to improve its performance. Additionally, Okolo et al. [44] employed eleven CNN models to classify chest radiograph images as belonging to healthy persons, people with COVID-19, or people with viral pneumonia. They analyzed three distinct improvements to modify the frameworks for the COVID-19 detection by expanding them with extra layers. All of the investigated networks are established frameworks that have been demonstrated to be effective in image detection and classification tasks. The proposed techniques were tested on a COVID-19 radiography database for all of the studied designs, with the EfficientNetB4 and Xception-based models providing the best classification results. Uddin et al. [45] suggested a CNN-based model for detecting COVID-19 from chest radiograph images, making the test more effective and trustworthy. The proposed model used a TL technique and a bespoke model to improve accuracy. The pre-trained CNN models, such as InceptionV3, MobileNetV2, ResNet50, and VGG16, were utilized to extract deep features. The categorization and classification accuracy were utilized as a criterion for gauging performance in this study. According to the results of this study, DL can detect SARS-CoV-2 from CXR images. Out of all of these TL models, InceptionV3 has achieved the highest accuracy.

Aside from ML approaches and DL models, previous research has also used hybrid models that combine both classic ML and DL-based methodologies. Sethy et al. [46] employed chest radiograph images to detect COVID-19 infected patients (deep feature and SVM-based approach) using a hybrid approach. For classification, SVM is used rather than a DL-based classifier because the latter requires a large training dataset. Deep features from the DL models' fully connected (FC) layers are obtained and fed into the SVM for COVID-19 categorization and classification. The distant chest radiograph data sources used in the technique are pneumonia, normal, and COVID-19. The method aids medical practitioners in distinguishing between healthy people, COVID-19, and pneumonia patients. The SVM algorithm is evaluated for COVID-19 detection using the attributes of 13 DL frameworks. The best results were from ResNet50 and SVM algorithms. Based on chest radiograph image data, Novitasari et al. [47] employed CNN architectures as feature extractors and the SVM as a classification tool to determine if the participants were normal, COVID-19 positive, or had pneumonia. The tests contrasted the kernel used, feature selection strategies, feature extraction frameworks, and different classes. The authors used resnet50, resnet18, resnet101, and googlenet TL approaches to separate three classes: normal, pneumonia, and COVID-19. They attained the maximum average accuracy using resnet50, resnet18, resnet101, and googlenet.

The studies mentioned above could be increased even further. Both the images sent to the model and the network's architecture are beneficial in diagnosing COVID-19

infection. As seen above, traditional ML, CNN, transfer learning, and hybrid algorithms have all been employed to detect COVID-19. The primary goal of this study is to obtain satisfactory results in detecting COVID-19 instances while avoiding false positives. When the findings are analyzed, it is clear that the proposed method is effective and simple for COVID-19 detection.

## 3. Methodology

We proposed the CovidDetNet approach for effective and efficient detection and classification of COVID-19 using chest radiographs images. We accomplished a three-class classification (COVID-19, pneumonia, and normal) because its automatic prediction and detection can help doctors in rapid and in-time identification of COVID-19 patients to propose an appropriate treatment approach based on the cause of infection. The abstract view of the proposed approach is shown in Figure 1, which comprises five main steps. To run the proposed approach, we fed chest radiography images as input to the model. Furthermore, the input images in the datasets are 1024 × 1024 pixels in size. Next, to ensure uniformity and speed up the processing, we applied certain pre-processing to resize the input images to 256 × 256 pixels. Furthermore, a CovidDetNet architecture with nine convolution layers was developed and designed to classify images into three categories for the optimized setups, i.e., COVID-19, normal, and viral pneumonia. The dataset is divided into training and testing sets for all experiments. More specifically, we used 70% of the dataset for model training and 30% for testing. Finally, we evaluated the proposed model on the COVID-19 radiography dataset. The proposed model consists of ten learned layers, i.e., nine convolutional layers and one FC layer. The details about the proposed approach are elaborated below.

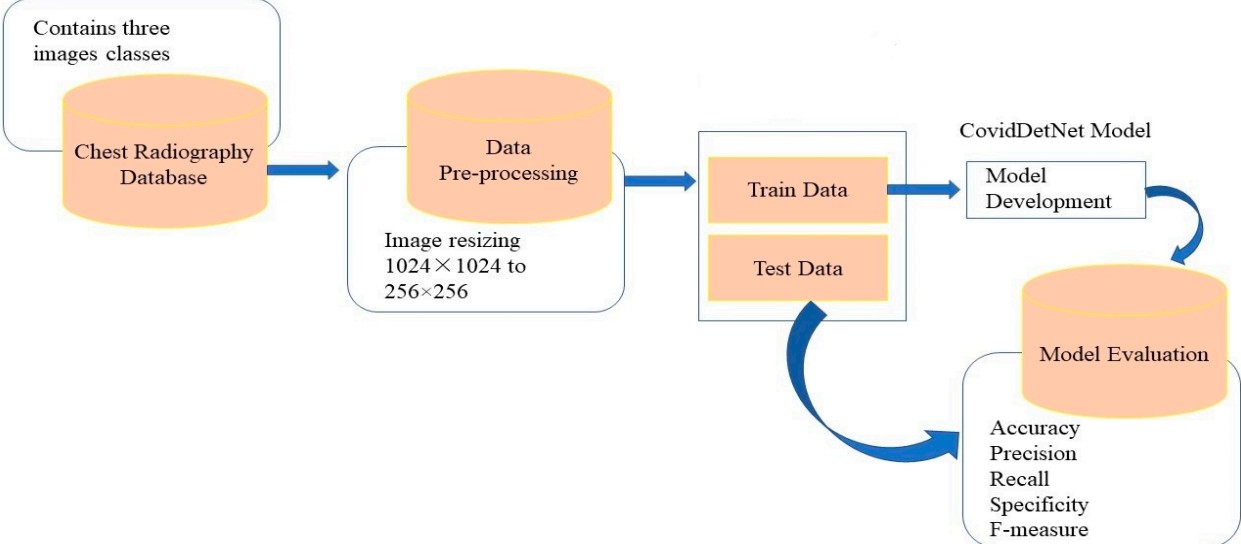

**Figure 1.** Abstract view of proposed CovidDetNet framework.

### 3.1. Motivations

DL algorithms may be a beneficial technique for detecting COVID-19 disease. DL techniques have been used to solve a variety of similar challenges, such as skin cancer classification [48], Parkinson and brain illness categorization [49], and pneumonia diagnosis using chest radiograph images [50]. In [51], the authors created a 121-layer DL-based model for pneumonia detection that produces the confidence score based on an input chest radiograph image. The model has been trained on millions of images and consistently outperforms practicing radiologists. It can detect 14 pathological disorders in chest radiographs with slight modifications using the most widely available chest radiograph dataset [52]. As COVID-19 has symptoms similar to severe pneumonia, therefore, inspired by the success of DL-based architecture in pneumonia detection using chest radiograph

images, we proposed a CovDetNet model for COVID-19 detection and classification in this work. The goal of this research is to propose a DL model that can effectively detect COVID-19 using chest radiographs with higher accuracy.

The proposed model's depth and input image resolution are based on the following facts: it is widely assumed that a deeper DL-based model captures more complicated and vital deep features and increases the network's classification performance. Furthermore, many DL-based models utilized depth scaling to improve their accuracy [53,54]. However, as the depth of the network expands, so does the computational complexity, which does not ensure that accuracy will improve in all circumstances. Therefore, DL-based models collect more detailed features with high-resolution input images to attain improved performance. DL models accept images with various resolutions, such as $224 \times 224$, $227 \times 227$, and $299 \times 299$, but models with higher resolution tend to perform better in recognition [55]. Similarly, the proposed model has ten layers and can process images with a resolution of $256 \times 256$ pixels. The CoviDetNet architecture and the size of the input image are chosen based on the present computing machine requirements.

Furthermore, we used both cross-channel [56] and batch [57] normalization (BN) layers in the proposed architecture. Cross-channel normalization layer because cross channel normalization improves generalization and reduces top-1 and top-5 error rates [56]. Batch normalization accelerates the model's training by reducing internal covariate shifting [57]. Batch normalization can stabilize DL models while maintaining data dispersion (data distribution). The input distribution varies as we train the CovidDetNet DL model, making the model train slower (internal covariate). We employ batch normalization to retain the same data distribution and deal with covariate shifting by normalizing the results (mean = 0, standard dev = 1). Furthermore, batch normalization normalizes each feature such that its significance is maintained, although some classification features have a greater numerical value than others. As a result, the proposed network will be unbiased (to higher-value classification features). Furthermore, in contrast to a network that does not utilize batch normalization, the model that uses this technique is trained faster and has higher accuracy.

Additionally, we used both LeakyRelu [58] and ReLu [59] activation functions in the proposed CovidDetNet model. Because ReLu is more computationally efficient, it only needs to select max (0, x) and not execute expensive exponential calculations. It indicates that ReLu neurons have zero derivatives for all negative inputs (output zero for negative values). Furthermore, a value of 0 indicates that the network will run faster on the negative axis. Moreover, we used the LeakyRelu [58] activation function in the first five layers to confound the dying ReLu issue. The ReLu activation procedure is extensively employed between layers to enhance non-linearity and handle more complicated information. However, as ReLu neurons have zero derivatives for all negative inputs, as a result of the network's weights continuously resulting in negative inputs to a ReLu neuron, that neuron is effectively not participating in the network's training (i.e., neurons die), and the problem is referred to as the dying ReLu issue. The DL network will stop learning in a dying ReLu issue. We used a leaky ReLu in the proposed CovidDetNet approach to overcome this issue. The Leaky ReLu activation process allows for a minor (non-zero) gradient whenever the unit is inactive. As a result, it continues to learn without coming to a halt, i.e., reaching a dead end. Furthermore, Max pooling is chosen in our proposed study because it preserves the most prominent characteristics of the feature maps, resulting in sharp classification features. Max-pooling with a filter size of $3 \times 3$, a stride of [2 2], and padding of [0 0 0 0] is used for downsampling. A global average pooling is used at the end of the structure to convert each feature map into a single value.

We believe that the proposed CovidDetNet architecture works well in classifying and identifying COVID-19 using chest radiographs images. It combines the advantages of the ReLu activation function, leaky ReLu activation function, batch normalization, and cross channel normalization layers to improve the performance of the proposed CovidDetNet in identifying and classifying COVID-19. Furthermore, the proposed model uses a high-

resolution image of 256 × 256 and is deep enough to capture more complex and essential features. Below, we explain the architecture of the proposed CovidDecNet approach.

### 3.2. CovidDetNet Architecture Details

The architecture of the proposed CovidDetNet model is depicted in Figure 2. As illustrated in Table 1, the CovidDetNet model has ten learnable layers: nine convolutional layers and one FC layer. The input layer in the proposed model is the initial layer, and it accepts 256 × 256 input images for processing. The architecture has a total of thirty-one layers, including two cross-channel normalization layers, three maximum pooling layers (to reduce network size), four batch normalization layers, three clipped ReLu layers, five leaky ReLu layers, one global average pooling layer, one Softmax layer, and one classification layer. In the proposed CovidDetNet model, the leaky ReLu (nonlinear activation function) is utilized after the first five convolutional layers. In contrast, the ReLu activation is employed after the last four convolutional layers. The first two layers of architecture are followed by the cross-channel sectional normalization layer, whereas batch normalization layers follow the last four convolutional layers. Maximum pooling layers follow the first two and fifth convolutional layers. The output of the last FC layer is provided as an input to 3-way Softmax in the case of COVID-19 classification and detection (three-class classification).

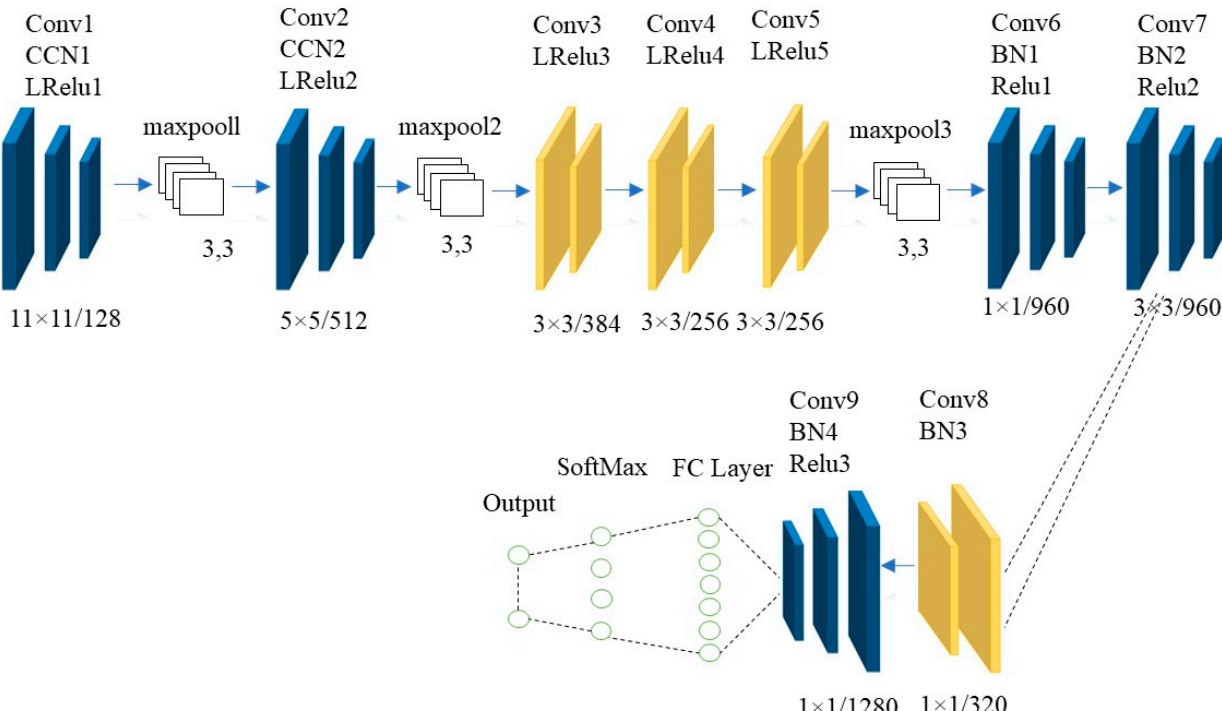

**Figure 2.** The architecture of the proposed CovidDetNet model.

The first convolution layer extracts feature from the 256 × 256 input chest radiograph image by using 128 filters of size 3 × 3 with a 4-pixel shift (stride) at a time. After employing normalization and maximum pooling operations, the output feature of the initial convolutional layer is provided into the second convolutional layer. The next convolutional layer uses 512 filters of size 3 × 3 having a stride value of 1-pixel at a time. The inputs are filtered by the third convolutional layer, employing 384 filters of size 3 × 3 with a stride value of 1 pixel. Pooling layers do not follow the subsequent two convolutional layers, the fourth and fifth. The fourth and fifth convolutional layers apply 256 filters of size 3 × 3 to the input feature map with a default stride of 1 pixel. The sixth convolutional layer applies 960 filters of size 1 × 1, whereas the seventh convolutional layer applies 960 filters of size 3 × 3. The eighth convolutional layer applies 320 filters of size 1 × 1, and the final (ninth) convolutional layer applies 1280 filters of size 3 × 3.

**Table 1.** CovidDetNet architecture parameters.

| S. No | Classification Layers | Kernels | Size | Padding | Strides |
|---|---|---|---|---|---|
| 1 | Input | | | | |
| 2 | Convolutional-layer-1 (LeakyRelu + Cross channel Normalization) | 128 | $11 \times 11$ | $0 \times 0$ | $4 \times 4$ |
| 3 | Max pooling | | $3 \times 3$ | $0 \times 0$ | $2 \times 2$ |
| 4 | Convolutional-2 (LeakyRelu + Cross channel Normalization) | 512 | $5 \times 5$ | $2 \times 2$ | $1 \times 1$ |
| 5 | Max pooling | | $3 \times 3$ | $0 \times 0$ | $2 \times 2$ |
| 6 | Convolutional-3 (LeakyRelu) | 384 | $3 \times 3$ | $1 \times 1$ | $1 \times 1$ |
| 7 | Convolutional-4 (LeakyRelu) | 256 | $3 \times 3$ | $1 \times 1$ | $1 \times 1$ |
| 8 | Convolutional-5 (LeakyRelu) | 256 | $3 \times 3$ | $1 \times 1$ | $1 \times 1$ |
| 9 | Max pooling | | $3 \times 3$ | $0 \times 0$ | $2 \times 2$ |
| 10 | Convolutional-6 (Batch normalization + ReLu) | 960 | $1 \times 1$ | same | same |
| 11 | Convolutional-7 (Batch normalization + ReLu) | 960 | $3 \times 3$ | same | same |
| 12 | Convolutional-8 (Batch normalization + ReLu) | 320 | $1 \times 1$ | same | same |
| 13 | Convolutional-9 (Batch normalization + ReLu) | 1280 | $1 \times 1$ | same | same |
| 14 | Average pooling layer | | | | |
| 15 | FC layer | | | | |
| 16 | Softmax | | | | |
| 17 | Classification | | | | |

### 3.3. Hyperparameters Settings

We employed a grid search technique to identify the optimal hyperparameters (which give high accuracy and less error) for the proposed CovidDetNet model. Given the numerous choices for layer types, numbers, and parameters, we chose to test and analyze the performance of the proposed CovidDetNet model with as few layers as possible, i.e., using only ten layers. After some preliminary trials on a smaller dataset, the proposed technique hyperparameters and extra layers are selected. Table 2 shows the details of the shortlisted parameters for the proposed CovidDetNet approach. We employed stochastic gradient descent (SGD) to train the proposed CovidDetNet method, using a final learning rate and minibatch of 0.001 and 10 images, respectively. The CovidDetNet classifier is trained over 60-epochs for COVID-19 detection (three-class classification) using chest radiograph images.

**Table 2.** Selected parameters for CovidDetNet classifier.

| CovidDetNet Parameters | Values Given |
|---|---|
| Learning rate | 0.001 |
| Validation frequency | 30 |
| Optimization algorithm | SGDM |
| Shuffle | Every epoch |
| Iterations per epoch | 82 |
| Maximum epochs | 60 |
| Verbose | false |
| Activation function | Leaky ReLu + ReLu |

## 4. Results

This section goes through the findings of a series of experiments conducted to assess the CovidDetNet model's performance. Furthermore, the section elaborates on further information about the datasets used to assess and evaluate the CovidDetNet model's performance, specifically the COVID-19 radiography database.

### 4.1. Datasets

We employed the COVID-19 radiography images dataset [60,61] for the proposed CovidDetNet approach, produced by a group of academics from the University of Dhaka, Bangladesh and Qatar University, Qatar, to detect COVID-19. This COVID-19 chest radiograph image database is developed and created in collaboration with medical doctors con-

taining COVID-19 positive samples, pneumonia, and healthy samples. The COVID-19 chest radiograph images database, which was recently released, includes 3616 chest radiographs of COVID-19 infected people, 10,192 chest radiograph images of healthy people, 6012 Lung Opacity, and 1345 images of pneumonia. The images in the dataset are 1024 × 1024 pixels in size. The images were resized to fit the need of each model. It is a standard Kaggle dataset that is freely available for research purposes. This dataset's radiograph images are grayscale and have the exact dimensions. A few example image samples from the SIRM dataset are displayed in Figure 3, whereas Table 3 provides statistical information about the dataset.

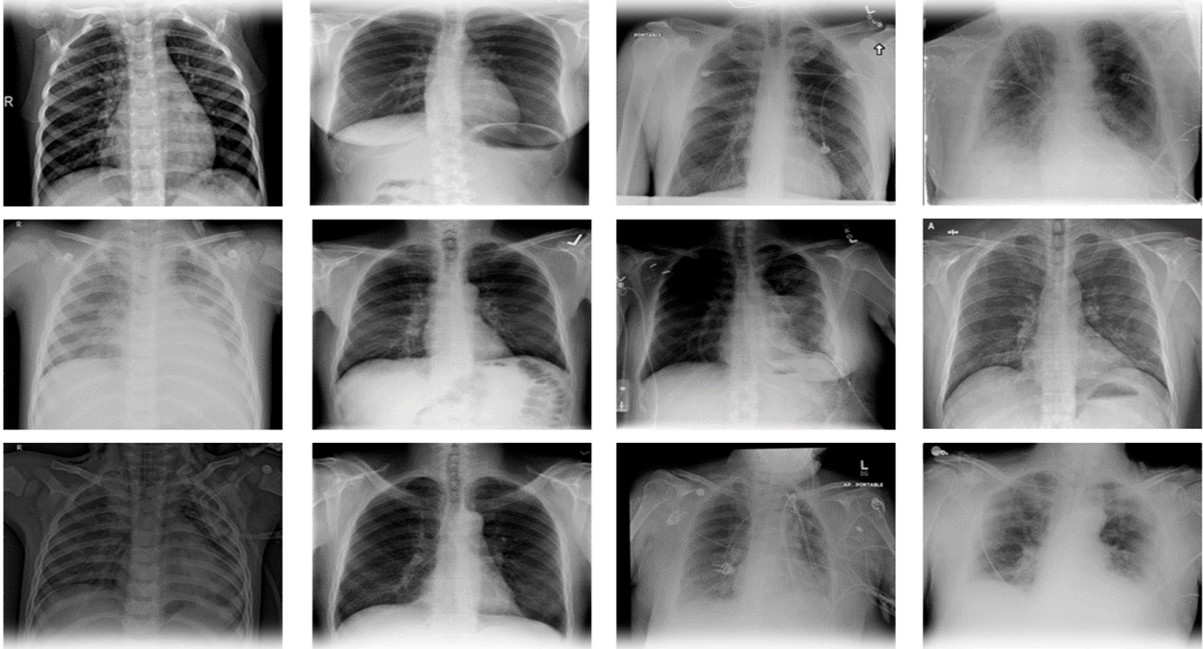

**Figure 3.** COVID-19 radiography database image samples, the first column have viral pneumonia, the second column has Normal, and the third column has lung obesity, whereas, the fourth column has COVID-19 chest radiograph images.

**Table 3.** Datasets details.

| Dataset | Images Type | Images Collections | | | Bit Depth | Format |
|---------|-------------|--------------------|--|--|-----------|--------|
| Chest Radiography Database | Chest Radiographs | COVID 3616 | Normal 10,192 | Pneumonia 1345 | 8 | PNG |

### 4.2. CovidDetNet Evaluation Metrics

The evaluation metrics such as accuracy, sensitivity (recall), precision, specificity, F1_score, and kappa are computed to validate the accurate performance of the proposed CovidDetNet approach, such that:

$$\text{Accuracy} = (\text{TN} + \text{TP})/\text{TS} \tag{1}$$

$$\text{Precision} = \frac{\text{TP}}{\text{TP} + \text{FP}} \tag{2}$$

$$\text{Sensitivity (recall)} = \frac{\text{TP}}{\text{TP} + \text{FN}} \tag{3}$$

$$\text{Specificity} = \frac{\text{TN}}{\text{TN} + \text{FP}} \tag{4}$$

$$F1\_score = 2 \cdot \frac{Precision \times Recall}{Precision + Recall} \tag{5}$$

$$Kappa = (p0 - pe)/1 - pe \tag{6}$$

where TP, TS, FP, TN, FN, p0, and pe are true positive, total samples, false positive, true negative, false negative, the proportion of cases correctly classified samples, and expected proportion of cases correctly classified by chance, respectively.

### 4.3. CovidDetNet Experimental Setup

Each experiment is performed on a laptop machine equipped with Intel (R) Core (TM) i5-5200U CPU and 8 GB RAM. To complete the research study, we used the R2020a version of MATLAB. For all experiments, the datasets are separated into training and testing datasets. The performance of a novel CovidDetNet framework for COVID-19 detection utilizing chest radiograph images is evaluated and analyzed by performing a series of DL experiments.

#### 4.3.1. Performance Evaluation

This experiment aims to validate the COVID-19 detection (three-class classification) performance of the proposed CovidDetNet framework using chest radiograph images. For this experiment, the dataset is divided into training and testing sets, i.e., we used 70% of the data for model training and 30% for testing. More specifically, we used all the 15,153 radiograph images (3616 data instances of COVID-19 patients, 1345 pneumonia radiographs, and 10,192 instances of healthy individuals) of the dataset named COVID-19 Radiography Database, where 10,606 images (2531 images of COVID-19 individuals, 941 pneumonia radiographs, and 7134 images of healthy individuals) were used for training and the remaining 4547 images (1085 images of COVID-19 individuals, 404 Pneumonia radiographs, and 3058 images of healthy individuals) for testing. The training set trains the proposed CovidDetNet framework for COVID-19 detection and classification with the same parameters mentioned in Table 2. The training of the proposed CovidDetNet model consumed 3853 min and 11 s for COVID-19 detection and classification. This time, however, is proportional to the maximum number of epochs and iterations per epoch. The total number of iterations in the training stage for CovidDetNet is 4920 (82 iterations per epoch), and the number of epochs is 60. At epoch 60, the model achieved an average validation accuracy of 98.40%. To assess the training performance of our approach we have shown accuracy and loss in Figures 4 and 5.

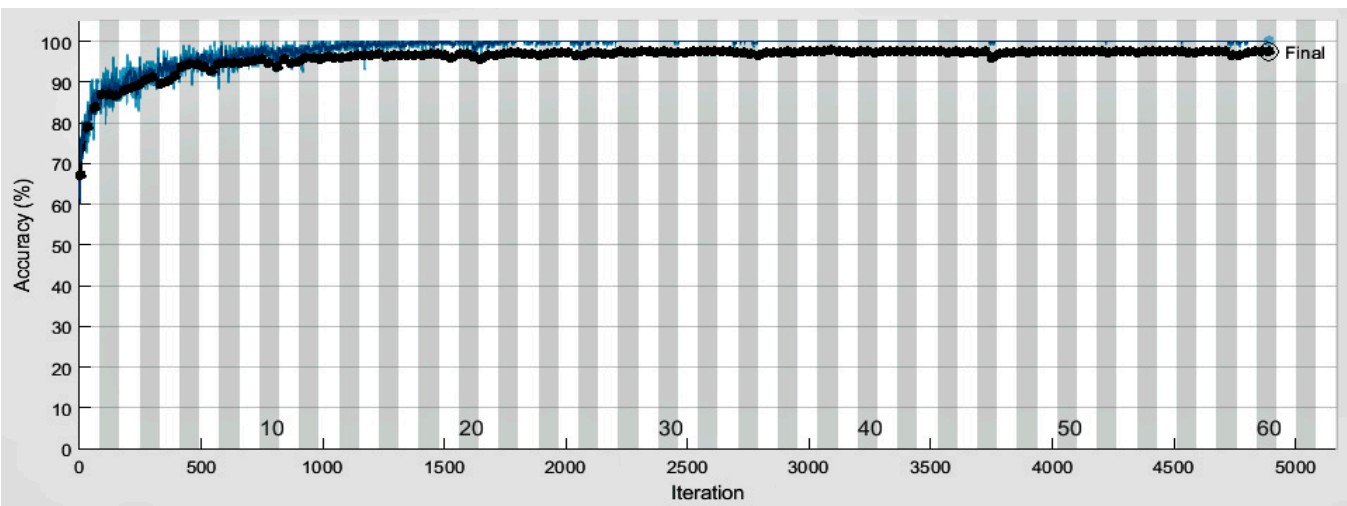

**Figure 4.** Training and validation accuracy of CovidDetNet model (black line shows testing accuracy whereas blue line shows training accuracy).

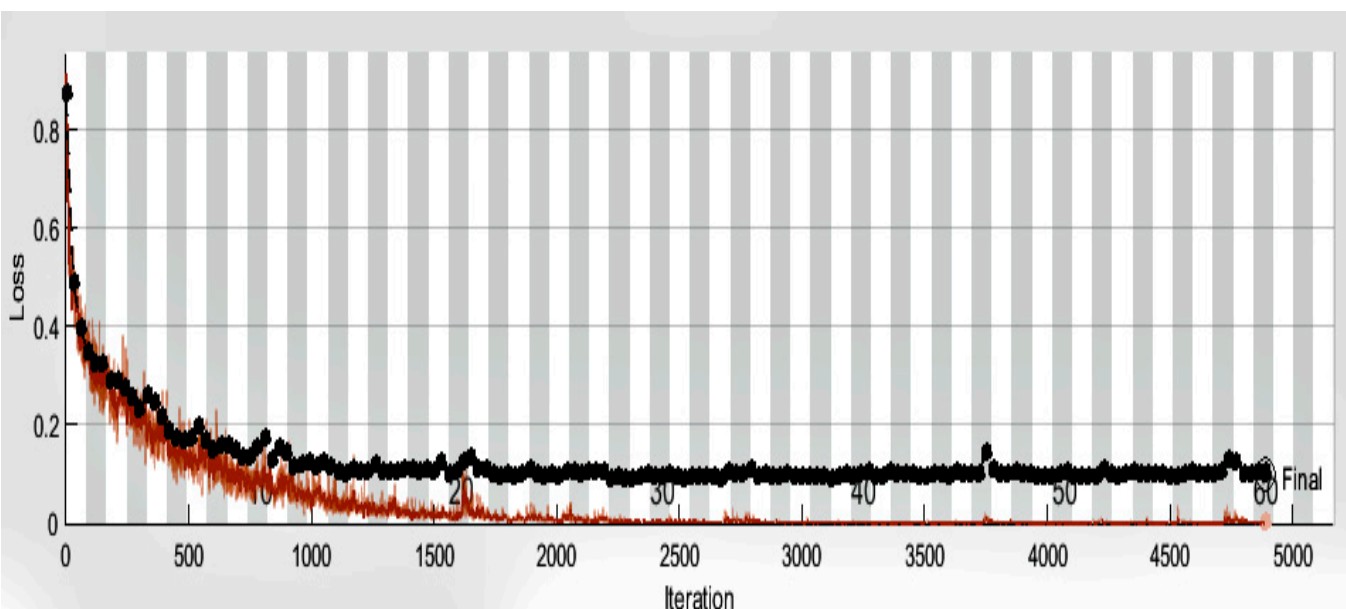

**Figure 5.** Training and validation loss of proposed CovidDetNet framework (black line shows testing loss whereas red line shows training loss).

The loss function indicates how well the framework can predict the dataset. The loss and accuracy of our model after epoch 47 approximately remains the same, which shows that our model is predicting COVID-19 with higher accuracy even at lower epochs than 60. The training and validation process of the proposed CovidDetNet approach is shown in Figure 4 and the confusion matrix for the testing phase of the CovidDetNet framework for COVID-19 infection detection is shown in Table 4. The proposed CovidDetNet architecture misclassified 109 radiographs out of 4547, of those, 48 COVID-19 images, 50 images for normal, and 11 images for viral pneumonia. The proposed CovidDetNet framework has higher TN and TP values, as well as lower FN and FP values, as shown by the confusion matrix. As a result, the suggested method is capable of accurately classifying COVID-19 situations.

**Table 4.** Confusion matrix of the proposed CovidDetNet model.

| | | Predicted Class | | |
|---|---|---|---|---|
| | | COVID-19 | Normal | Viral Pneumonia |
| Actual class | COVID-19 | 1037 | 46 | 2 |
| | Normal | 36 | 3008 | 14 |
| | Viral Pneumonia | 3 | 8 | 393 |

In addition, the training epochs in the experiments are changed from 20 to 60 with a step size of 10 to ensure the validity of the obtained results. Table 5 shows the accuracy, precision, recall, F1-score, and kappa values obtained by varying different values of epochs. The best results in terms of all performance measures are obtained with 60 training epochs. It is concluded that the accuracy of the proposed approach increases gradually with the increase in the number of epochs. The proposed CovidDetNet approach attained the average accuracy, precision, recall, specificity, F1-score, and Kappa index of 98.40%, 97.0%, 96.66%, 97.06%, 96.82%, and 95%, demonstrating its reliability in COVID-19 detection, as shown in Table 5. Figure 6 shows the receiver operating characteristic (ROC) curve of the proposed CovidDetNet framework. The area under the curve is 0.9955. Similarly, Figure 7 demonstrates the box plot of the proposed CovidDetNet approach with various values of epochs. It is demonstrated that increasing the size of epochs improves the values of various evaluation matrices, as shown in Table 5.

**Table 5.** Accuracies (in %) on different epochs.

| Epochs | Accuracy | Precision | Recall | F1-Score | Kappa |
|---|---|---|---|---|---|
| 20 | 95.72 | 92.33 | 93.33 | 92.82 | 95.0 |
| 30 | 98.21 | 96.66 | 96.66 | 96.66 | 95.0 |
| 40 | 98.25 | 96.66 | 96.66 | 96.66 | 95.0 |
| 50 | 98.3 | 97.00 | 96.66 | 96.82 | 95.0 |
| 60 | 98.40 | 97.00 | 96.66 | 96.82 | 95.0 |
| Average | 97.77 | 95.93 | 95.99 | 95.96 | 95.0 |

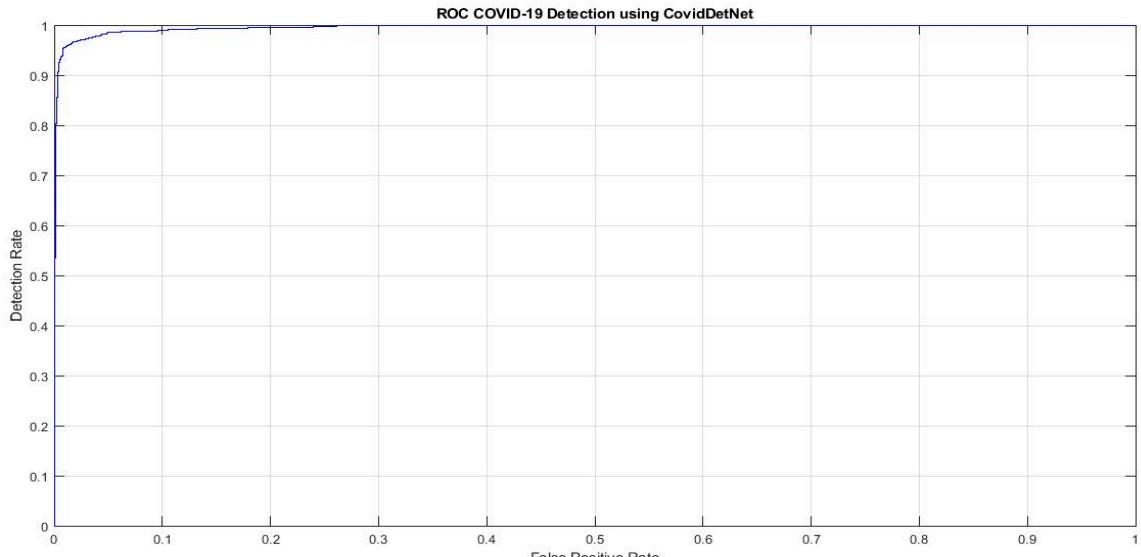

**Figure 6.** ROC of CovidDetNet framework for COVID-19 detection, the y-axis represents true-positive rate and false-positive rate.

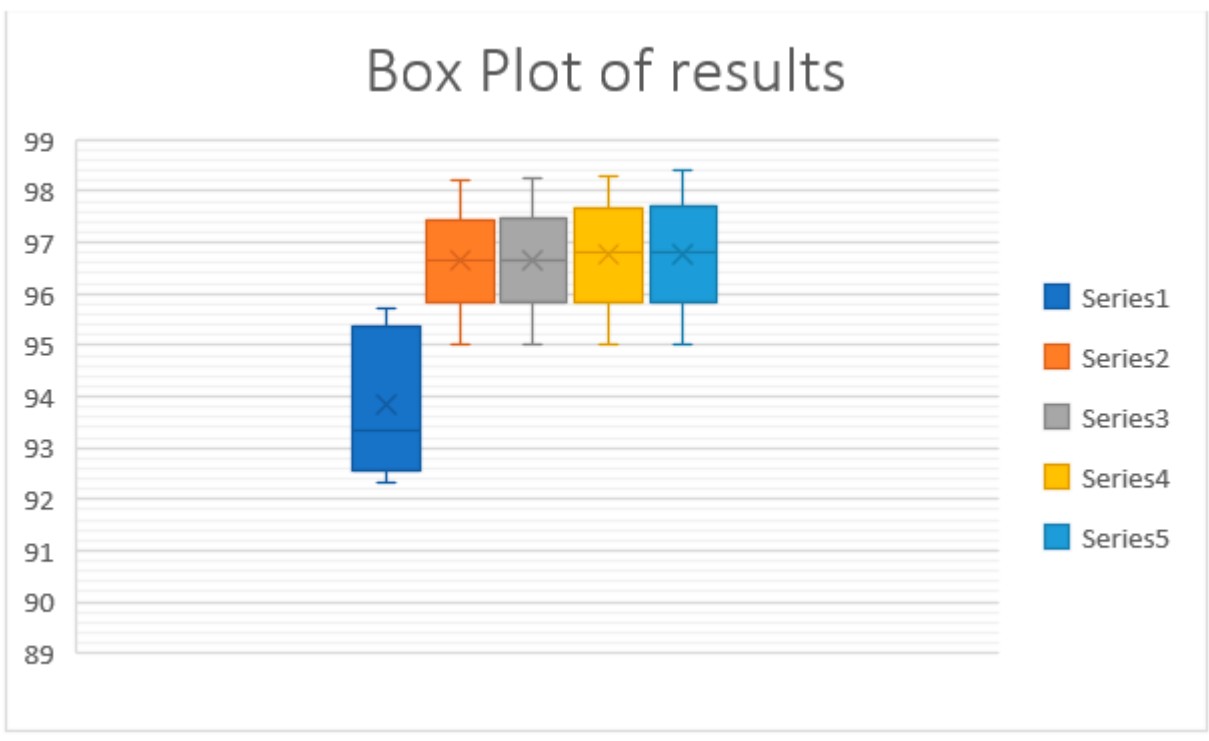

**Figure 7.** Box plot of results, Series1, Series2, Series3, Series4, and Series5 represents results for epochs 20, epochs 30, epochs 40, epochs 50, and epochs 60, respectively.

This demonstrates the impact of our proposed method on improving COVID-19 classification accuracy from chest radiographs. These results are because our proposed model successfully extracts the most discriminative, robust, and detailed deep features to represent the chest radiograph image for accurate and reliable classification. As it does not have a separate feature extraction stage, the suggested solution is straightforward to apply because of its end-to-end learning architecture.

### 4.3.2. Performance Evaluation on Train/Validation/Test Split

This experiment aims to validate further the COVID-19 detection (three-class classification) performance of the proposed CovidDetNet framework using chest radiograph images. For this experiment, the dataset is divided into three parts, i.e., training, validation, and testing sets. We used 70% of the input data for training, 10% for validation, and 20% for testing the proposed CovidDetNet approach. More specifically, we used all the 15,153 radiograph images (3616 data instances of COVID-19 patients, 1345 pneumonia radiographs, and 10,192 instances of healthy individuals) of the dataset named COVID-19 Radiography database, where 10,606 images (2531 images of COVID-19 individuals, 941 pneumonia radiographs, and 7134 images of healthy individuals) are used for training. In contrast, 1515 radiograph images (362 images of COVID-19 individuals, 134 pneumonia radiographs, and 1019 radiograph images of healthy individuals) are used for validation. The remaining 3032 images (723 images of COVID-19 individuals, 270 pneumonia radiographs, and 2039 images of healthy individuals) are used for testing. The training set trains the proposed CovidDetNet framework for COVID-19 detection and classification with the same parameters mentioned in Table 2. The training of the proposed CovidDetNet model consumed 3510 min and 36 s for COVID-19 detection and classification. This time, however, is proportional to the maximum number of epochs and iterations per epoch. The total number of iterations in the training stage for CovidDetNet is 4920 (82 iterations per epoch), and the number of epochs is 60. At epoch 60, the model achieved average validation and testing accuracies of 98.41% and 98.24, respectively. The validation and testing confusion matrices of the proposed CovidDetNet approach are shown in Tables 6 and 7, respectively. The results obtained from the train, validation, and test splits are shown in Table 8.

**Table 6.** Validation confusion matrix of the proposed CovidDetNet model.

| **Predicted Class** | | | | |
|---|---|---|---|---|
| | | COVID-19 | Normal | Viral Pneumonia |
| Actual class | COVID-19 | 350 | 12 | 0 |
| | Normal | 14 | 1001 | 4 |
| | Viral Pneumonia | 2 | 4 | 128 |

**Table 7.** Testing confusion matrix of the proposed CovidDetNet model.

| **Predicted Class** | | | | |
|---|---|---|---|---|
| | | COVID-19 | Normal | Viral Pneumonia |
| Actual class | COVID-19 | 688 | 33 | 2 |
| | Normal | 25 | 2004 | 10 |
| | Viral Pneumonia | 3 | 7 | 260 |

**Table 8.** Detailed results obtained with train, validate, and test split.

| | **Accuracy** | **Precision** | **Recall** | **F1-Score** | **Kappa** |
|---|---|---|---|---|---|
| Training | 100 | 100 | 100 | 100 | 100 |
| Validation | 98.41 | 97.0 | 97.0 | 97.0 | 94.5 |
| Testing | 98.24 | 96.33 | 96.66 | 96.50 | 94.5 |

### 4.3.3. Comparison with a Hybrid Approach

A hybrid experiment for COVID-19 detection is elaborated in this section to examine the effectiveness of the proposed CovidDetNet classifier. It is claimed that putting an SVM classifier at the top of the network instead of traditional deep CNN increases classification performance significantly [32]. As a result, we devised a hybrid strategy in which we extracted in-depth features using the eight most well-known deep CNNs and then used these features as inputs to train SVM with the linear kernel (decision function). The values of Gamma and C hyperparameters are set to 0.1 and 1.0, respectively, because these settings yielded the maximum performance. In the proposed research, we employed Alexnet [56], Resnet18 [62], Squeeznet [63], Darknet19 [64], Googlenet [65], Shufflenet [66], Resnet50 [62], Mobilnetv2 [67], and Inceptionv3 DL-based classification approaches.

For this experiment, the dataset is divided into training and testing sets, i.e., we used 70% of the data for model training and 30% for testing. More specifically, we used all the 15,153 radiograph images (3616 data instances of COVID-19 patients, 1345 pneumonia radiographs, and 10,192 instances of healthy individuals) of the dataset named COVID-19 Radiography Database, where 10,606 images (2531 images of COVID-19 individuals, 941 Pneumonia radiographs, and 7134 images of healthy individuals) were used for training and remaining 4547 images (1085 images of COVID-19 individuals, 404 pneumonia radiographs, and 3058 images of healthy individuals) for testing. We used the same experimental setup (hyperparameters values are selected using the same approach as the proposed method) to train these models as mentioned in Table 2. Table 9 contains information about these deep feature extractor models. As different models need input images of different sizes, such as mobilenetv2 accepting 224-by-224 input images, whereas darknet19 requires 256-by-256 input images. The dataset images are automatically resized using augmented image data repositories before being inserted into the network for feature extraction. We employed activations on deeper layers (last fully connected or global average pooling layer) because they include more high-level information than earlier layers; for example, we applied activations on the fc8 layer, the last layer (FC layer) in Alexnet. These layers pool the input features overall spatial locations after applying the activation functions to provide distinct features (i.e., Shufflenet gives 1000 features in total). Table 10 shows the classification results of deep features and the SVM technique. Compared to CovidDetNet, deep features of all twelve networks and the SVM technique produced poorer accuracy outcomes. Based on the experimental results, the proposed CovidDetNet approach outperforms the other eight hybrid models, reaching a COVID-19 detection accuracy of 98.40%. The Resnet18 model had the second-highest accuracy of 97.14%, while Inceptionv3 had the lowest accuracy of 95.05%. Resnet18 outperforms other models with a recall of 94.66% in terms of true positive rate. It is to be notice that the accuracy of all hybrid comparative models is greater than 95%. Resnet18 attained the second-highest accuracy because it uses batch normalization to reduce generalization error. The proposed CovidDetNet approach successfully extracts more distinguishing features from the chest radiograph images, which is why the proposed approach achieved better results in detecting and identifying COVID-19. We used small filters with $3 \times 3$ and $1 \times 1$, which ensured the extraction of more detailed and robust features. The proposed model's batch normalization technique standardizes the inputs to a layer for each mini-batch, offers regularization, and decreases the generalization error.

**Table 9.** Deep features extractor models details.

| S. No | CNN Architecture | Input Size | Depth | Activation Layer Name | Features |
|---|---|---|---|---|---|
| 1 | Alexnet | 227 × 227 | 8 | fc8 | 1000 |
| 2 | Resnet18 | 224 × 224 | 18 | pool5 | 512 |
| 3 | Squeezenet | 227 × 227 | 18 | pool10 | 1000 |
| 4 | Darknet19 | 256 × 256 | 19 | avg1 | 1000 |
| 5 | Googlenet | 224 × 224 | 22 | pool5-drop_7 × 7_s1 | 1024 |
| 6 | Shufflenet | 224 × 224 | 50 | node_200 | 544 |
| 7 | Resnet50 | 224 × 224 | 50 | avg_pool | 2048 |
| 8 | Mobilnetv2 | 224 × 224 | 53 | global_average_pooling2d_1 | 1280 |
| 9 | Inceptionv3 | 299 × 299 | 48 | avg_pool | 2048 |

**Table 10.** COVID-19 detection using chest radiographs comparison with hybrid models.

| Model | Accuracy | Precision | Recall | Specificity | F-Measure |
|---|---|---|---|---|---|
| Alexnet | 96.13 | 93 | 92.33 | 97.07 | 92.66 |
| Resnet18 | 97.14 | 94.66 | 94.66 | 97.43 | 94.66 |
| Squeezenet | 95.99 | 93 | 93.33 | 96.87 | 93.16 |
| Darknet19 | 96.87 | 93.33 | 94.33 | 97.43 | 93.83 |
| Googlenet | 95.82 | 92 | 93 | 96.83 | 93 |
| shufflenet | 96.09 | 92.33 | 93.33 | 97.61 | 92.83 |
| Mobilenetv2 | 96.29 | 93.33 | 93.33 | 96.81 | 93.33 |
| Inceptionv3 | 95.05 | 90.33 | 92.0 | 95.76 | 91.16 |
| Proposed CovidDetNet | 98.40 | 97 | 96.66 | 97.06 | 96.82 |

### 4.3.4. COVID-19 Detection Comparison with State-of-the-Art Methods

This experiment aims to recover the performance of the proposed CovidDetNet framework in identifying and classifying COVID-19 from chest radiograph images compared to existing state-of-the-art deep learning approaches in the literature. We compared the proposed classification model performance to various approaches [43,44,68,69]. Prateek et al. [68] proposed an automated diagnostic method for COVID-19 detection and classification using the DL model on chest radiograph images. The Inception-V3 model, with node dropping, flattening, dense layer, and normalization, was used to automatically present a transfer learning-based algorithm for detecting COVID-19 from chest radiographs. Three separate COVID-19 X-rays datasets with three classes (COVID-19, normal, and pneumonia) were used to test the model's effectiveness. The suggested framework has achieved a maximum accuracy of 97.7%. Similarly, Aayush et al. presented the SARS-Net DL model, an automatic method for COVID-19 detection using graph convolutional networks and CNNs for identifying anomalies in a patient's chest radiographs for the presence of COVID-19 virus [69]. The suggested model also outperformed the state-of-the-art methodologies previously discussed. On the validation set, the suggested model had an accuracy of 97.60% and a sensitivity of 92.90%. Gabriel et al. [43] evaluated eleven deep CNN frameworks for the classification of chest X-rays into healthy people, people with COVID-19, and pneumonia. They focus on three distinct adjustments to adapt the designs for the COVID-19 classification task by adding new layers to them. The proposed techniques were tested and analyzed on a chest X-ray images dataset, with the best-performing setup achieving the maximum classification accuracy of 98.04% and the highest F1-score of 98.22%. Azhar et al. [44] introduced a CNN framework for detecting COVID-19 using chest radiographs that are faster and more reliable. For feature extraction, a CNN approach was utilized. Four convolutional layers, three MaxPooling layers, one flattened layer, and two thick layers with a ReLu activation function make up the model. In the final layers, pretrained models such as InceptionV3, Resnet50, MobileNetV2, and VGG16 were employed with some modifications. The created model had a validation accuracy of 98%.

The comparison results are shown in Table 11, elaborating the success of the proposed CovidDetNet model in identifying COVID-19 from chest radiograph images compared to existing alternatives. It is important to mention that the proposed model outperformed the approaches [43,44] using the same dataset (COVID-19 radiography database) for COVID-19 detection. It is worth noting that these approaches are more computationally challenging than the proposed approach because they use deeper models, leading to overfitting. The proposed CovidDetNet model, on the other hand, is ten layers deep and uses modest $1 \times 1$ filters to extract in-depth high-level and more complex features for COVID-19 identification.

**Table 11.** COVID-19 detection using chest radiographs comparison with existing methods.

| Work | Method | Dataset | Date | Accuracy |
|------|--------|---------|------|----------|
| Prateek et al. [68] | Inception-V3 with flattening, node dropping, normalization, and dense layer | Chest X-ray (COVID-19 and Pneumonia) 2020 | 2021 | 97.7 |
| Aayush et al. [69] | SARS-Net | COVIDx | 2021 | 97.60 |
| Gabriel et al. [43] | Round-off fine-tuning | COVID-19 radiography database | 2021 | 98.04 |
| Azhar et al. [44] | Custom CNN | COVID-19 radiography database | 2021 | 98 |
| Proposed approach | CovidDetNet model | COVID-19 radiography database | 2022 | 98.40 |

## 5. Discussion and Future Research Direction

In this research, we have adopted a model with higher accuracy (98.40) than competing models in detecting COVID-19 from chest radiography. The training and testing accuracy of the model proliferates after each epoch, as shown in Figure 4, and the training and testing loss decrease gradually, as depicted in Figure 5. Although the prop approach yielded promising results, we recognized several limitations and made some recommendations for future research. Due to the unavailability of the research datasets that could be used as a baseline to investigate the severity level of COVID-19 infection, the proposed approach was unable to categorize the various stages of COVID-19 infection, such as pre-symptomatic, asymptomatic, moderate, and severe. The proposed method does not reveal how well the system detects COVID-19 using other imaging modalities such as computerized tomography (CT scans) and electrocardiogram (ECG) trace images. We continuously split image data into a 70% training set and a 30% test set in the proposed approach. Alternative splits, on the other hand, may produce different results. In the future, we plan to conduct experiments using comparable large-scale datasets of chest radiographs, CT scans, and ECG trace images to assess the models' generalization ability by testing them on a range of large-scale datasets from diverse sources and images obtained by several machines. We want to use the same method to categorize the various stages of COVID-19 infection, such as pre-symptomatic, asymptomatic, mild, severe, and so on. According to the experimental results, the proposed CovidDetNet method is more accurate than PCR because PCR results rely heavily on sample collection timing, type, storage, handling, and processing. A false-negative result is possible if the sample is not properly obtained or if an individual is tested too early after exposure to the virus or too late in their infection. Therefore, in the future, we intend to provide further experimental evidence to compare the performance of the proposed system (CovidDetNet approach) with PCR and other manual COVID-19 testing methods to identify its performance in identifying COVID-19. The proposed CovidDetNet approach estimated accuracy might be biased or inflated because we used a dataset containing images with only three classes (normal, pneumonia, covid) for training and testing the proposed model. Therefore, in the future, we intend to use the proposed method in other COVID-19 datasets or other medical datasets with CT scans or chest radiographs to test the generalization ability of the proposed CovidDetNet model so that it can be used in practice to detect different diseases such as tuberculosis, breast cancer, and lung opacity, etc. Furthermore, the challenge faced by the machine and deep learning experts these days is the unavailability of relevant data. Currently, the dataset adopted for the proposed

approach contains only three types of images such as normal, COVID-19, and pneumonia. Therefore, to generalize the proposed CovidDetNet approach, we need to evaluate it with other types of chest radiograph disease images. For this purpose, in the future, we need to explore the existing benchmark dataset of chest radiographs containing breast cancer, tuberculosis, and lungs opacity together with the COVID-19 detection. This paper only focuses on the approaches that use deep learning to detect and identify COVID-19 from chest radiograph images. In the future, we are interested in conducting an experiment with expert radiologists and identifying the proposed approach's efficacy in identifying COVID-19. It is currently difficult to identify radiologist experts in the area where most of the authors reside. We claim that the proposed CovidDetNet approach can be used as a replacement for expert radiologists to timely identify COVID-19 and help stop the spread of the virus. Furthermore, to perfectly know the time gained comparing chest radiography with the proposed CovidDetNet approach, we need to experiment with the domain experts diagnosing COVID-19 from X-rays and compare the results and time taken with the proposed CovidDetNet approach. Additionally, to generalize the proposed CovidDetNet approach, we aim to test and validate the proposed approach with the emerging new variants of COVID-19 and record its performance and accuracy to identify the new types of COVID-19 virus. Additionally, we are interested to identify the performance of proposed approach on detecting & removing mask [70] and heart disease predication [71,72].

Furthermore, healthcare institutions in many countries are incorporating a large number of smart devices for combating the disease and obtaining information about its growth. In addition, blockchain and IoT are also assisting medical professionals in gaining valuable insights about behavior and symptoms. At the same time, physicians are using various IoT-enabled devices for remote monitoring of patients, considering that COVID-19 spreads faster than the average communicable disease. In particular, the Internet of Medical Things (IoMT) applications involve tracking medication orders, monitoring COVID-19 patients remotely, and incorporating wearables for transmitting healthcare information to the respective healthcare professionals. The healthcare sector is banking on the potential of IoMT technologies to collect, evaluate, and transmit healthcare information efficiently. IoT devices can pacify the diagnosis process of infectious diseases, which is essential in the case of COVID-19. IoT-enabled devices can capture body temperatures, collect samples from possible cases, and eliminate human intervention. Even during the quarantine period, IoT devices can remotely monitor patients, preventing further infection. Even IoT-based drones are being used for thermal imaging, disinfecting, medical purposes, surveillance, and announcements to draw the line of defense against COVID-19. Therefore, there is a rising demand to develop deep learning models based on IoT and blockchain technologies—since both have significantly leveraged the global healthcare industry—to detect COVID-19 infection early from the data gathered from IoT-enabled devices.

## 6. Conclusions

COVID-19 infection needs to be identified and detected early to prevent the infection from spreading to others. This research study proposed a novel CovidDetNet classification approach to efficiently and correctly identify COVID-19 using chest radiograph images. For this purpose, following image resizing, the resulting images are fed into a CovidDetNet model developed to detect COVID-19. The accuracy of 98.40% for COVID-19 detection has confirmed the superiority of our CovidDetNet model over other existing hybrid approaches. The results of our rigorous testing demonstrated that our proposed model outperforms other contemporary techniques. In the COVID-19 global pandemic, the proposed CovidDetNet classification and identification approach is expected to develop a mechanism for COVID-19 patients and reduce COVID-19 medical diagnosis workload and virus spread. Because the proposed CovidDetNet model has an end-to-end learning structure, it can detect COVID-19 from chest radiograph images automatically without the necessity for any manual feature extraction techniques. As a decision support system, a rapid and stable system aids expert radiographs in this way. The workload of radiologists can be reduced,

and misdiagnosis is avoided. Despite the success of the given method, different DL-based strategies for COVID-19 identification will be proposed in future studies to improve the CovidDetNet approach's performance further.

**Author Contributions:** N.U. developed the method; N.U., M.S.K., S.A. and J.A.K. performed the experiments and analysis, and M.A., D.A. and A.R. wrote the paper. All authors have read and agreed to the published version of the manuscript.

**Funding:** The authors are thankful to the Deanship of Scientific Research at Najran University for funding this work under the Research Collaboration Funding program grant code (NU/RC/SERC/11/8).

**Institutional Review Board Statement:** Not applicable.

**Informed Consent Statement:** Not applicable.

**Data Availability Statement:** The datasets used in this investigation are available on request from the corresponding author.

**Conflicts of Interest:** The authors declare no conflict of interest.

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
