# Peer review of "A Novel CovidDetNet Deep Learning Model for Effective COVID-19 Infection Detection Using Chest Radiograph Images"

_applsci, doi:10.3390/app12126269_

Round 1

Reviewer 1 Report

In this manuscript, the authors propose a novel deep network architecture dedicated to aid at the detection of Covid-19 in thoracic radiography images. The work is well-written and organized. The methodological aspects seem clear and reproducible. However, in order to aid the authors to improve the quality of their scientific communication, I suggest the following adjustments:

  1. Although many good works on deep learning demonstrated the feasibility of their results by performing one-shot learning, to be correct from a statistical learning point of view, it is not sufficient. Since you até using artificial neural networks, these models are all randomly initiated. Therefore, I suggest the authors to perform at least 25 runs and 10-fold cross validation. Afterwards, the authors should present the experimental results for their adopted metrics as tables filled with sample average and standard deviation. Additionally, they should present their experimental results as box plots or violin plots
  2. Regarding the metrics, I suggest to include the area under ROC curve and the Kappa index. Though Kappa has no clinical value, it is important to highlight differences among classification models from a statistics and computing point of view
  3. Considering your related works, I suggest the authors to include some works based on explicit attributes, especially texture features, due their importance to the clinical application and reception of the technology by clinicians. Furthermore, you will have other approaches to compare your model without focusing just on deep learning. 

Author Response

Dear Reviewer, 

Please find the attached detailed response to your comments. The Comments contains Figures, Table and some equations that why they are uploaded in the documents. I hope the response will satisfy your high-quality reviews. 

Kind Regards

Reviewer 2 Report

The paper presents a new method CovidDetNet for predicting if patients have COVID-19 by analyzing radiography using a Convolution Neural Network. The authors describe the method and present experiments comparing it to other state-of-the-art algorithms.

Major concerns:

1) The advantages of the proposed method are not clear. The work is a good contribution in the sense of using Machine Learning to predict a disease. However, the clinical advantages are not clear.

First of all, the authors claim that COVID-19 tests are slow, as they take hours for results, which brings the need for the proposed technique. However, it is not mentioned what is the time gain of doing a Radiography and having one specialist running a tool (CovidDetNet) to get the clinical results.

Another claim is that the proposed method is accurate. However, how does it compare to PCR or even over-the-counter COVID-19 tests accuracy, which gives results at home in a couple minutes?

Finally, the authors mention the cost as one important factor, claiming that COVID-19 tests cost about $60. But it does not take into account what is the cost of a radiography, and the time taken by a technician to analyze results using CovidDetNet.

Therefore, it is important to have these points more clear in the Abstract and Introduction.

2) The experimental designs needs improvement. Authors claim that hyperparameter setting was done in a "trial-and-error strategy", which does not seem appropriate. Since the authors have a dataset with 70% training and 30% test, you are essentially find parameters that give least test error. However it is not clear if this generalizes to unseen data. 

More work needs to be done, such as having a validation set, where parameters are selected independently of the test set. Alternatively, performing cross-fold validation.

In the same line of concern, it is not clear how the parameters of the methods being compared to were selected.

Minor concerns:

1) The introduction of pneumonia as one of the classification tasks is not clearly defined. I see that adding this data to training of the CNN would bring benefits similar to transfer learning, since it has probably some shared representation in inner layers of the neural network. However, pneumonia was not explored in the paper and this information is lost.

2) There are minor spelling error, such as "iinovation" on page 2 line 57 and "reflux" on page 3 line 128. 

Author Response

Dear Reviewer, 

Please find the below-detailed response to your comments.  I hope the response will satisfy your high-quality reviews. The corresponding changes are highlighted in the manuscript with a yellow background to make them prominent. The response is attached below for better readability. 

Kind Regards

---------------------------------------------------------------------------------

Reviewer 3 Report

Although results are interesting, the main issue of this study has to do with its design. Specifically, apparently only images from three classes are fed to the algorithm for training and testing (normal, pneumonia, covid). However, in everyday clinical practice chest images will arise from a multitude of conditions (as the authors indirectly acknowledge in section 3.1). The designed model's estimated accuracy is highly biased/inflated, as a result. 

So, the overall impression is that this model will never actually be used in practice.

Furthermore, there are clinical concerns as evolving variants do not affect the lungs but rather the upper respiratory system. This study, unfortunately, comes a bit late in this sense.

Author Response

Dear Reviewer, 

Please find the below-detailed response to your comments.  I hope the response will satisfy your high-quality reviews. The corresponding changes are highlighted in the manuscript with a yellow background to make them prominent. The response is attached for better readability.

Kind Regards

--------------------------------------------------------------------------------

Round 2

Reviewer 2 Report

1) Although the authors have made significant attempts to explain the advantages of the proposed work, there are still gaps that need to be explained.

First of all, authors mention that COVID-19 tests take hours to generate results, while their procedure takes 15 minutes. However, this does not take into account self-home kits, in which results are usually ready in 10 minutes, without the need of an expert to analyze results. The price comparison should also be done with respect to self-home kits, which are usually cheaper.

I also understand that authors have a future work designing experiments comparing the proposed approach to PCR and other COVID-19 test methods. However, the lack of references from literature on how accurate existing tests are reduces the significance and applicability of the proposed work.

2) I am still very concerned about the grid search strategy that authors have used. It is understandable that 10-fold CV is not feasible to the resources available. However, dividing the dataset into training / test and selecting hyperparameters that give smaller test error, you are essentially showing that your method has parameters that overfits to test set. If 10-Fold CV is not doable, please divide the dataset into 3 parts: training / validation / test set. Then, select hyperparameters to minimize the error on the validation set, and show the model performance comparing to baselines on the test set. Please report the hyperparameter selection grid in Appendix.

Author Response

Thank you, for the valuable comments that help us to improve the quality of the manuscript. The detailed reply to the comments is attached below. We are hopeful that this version of the manuscript will satisfy the needs of the esteemed reviewer. 

Kind Regards

Reviewer 3 Report

The authors have added comments indicating the existing inherent problems in this research. They have also revised according to other reviewers' comments. In this sense the revised version of the article is better than the previous one.

Author Response

Thank you very much for accepting the replies in response to your comments. We appreciate your dedication and commitment to research community. 

Kind Regards